

# The hidden anatomy of paranasal sinuses reveals biogeographically distinct morphotypes in the nine-banded armadillo (*Dasypus novemcinctus*)

Guillaume Billet[1], Lionel Hautier[2,3], Benoit de Thoisy[4,5] and Frédéric Delsuc[2]

[1] Sorbonne Universités, CR2P, UMR 7207, CNRS, Université Paris 06, Museum national d'Histoire naturelle, Paris, France
[2] Institut des Sciences de l'Evolution, UMR 5554, CNRS, IRD, EPHE, Université de Montpellier, Montpellier, France
[3] Mammal Section, Life Sciences, Vertebrate Division, The Natural History Museum, London, United Kingdom
[4] Institut Pasteur de la Guyane, Cayenne, French Guiana, France
[5] Association Kwata, Cayenne, French Guiana, France

Corresponding author
Guillaume Billet,
guillaume.billet@mnhn.fr

## ABSTRACT

**Background.** With their Pan-American distribution, long-nosed armadillos (genus *Dasypus*) constitute an understudied model for Neotropical biogeography. This genus currently comprises seven recognized species, the nine-banded armadillo (*D. novemcinctus*) having the widest distribution ranging from Northern Argentina to the South-Eastern US. With their broad diversity of habitats, nine-banded armadillos provide a useful model to explore the effects of climatic and biogeographic events on morphological diversity at a continental scale.

**Methods.** Based on a sample of 136 skulls of *Dasypus* spp. belonging to six species, including 112 specimens identified as *D. novemcinctus*, we studied the diversity and pattern of variation of paranasal cavities, which were reconstructed virtually using μCT-scanning or observed through bone transparency.

**Results.** Our qualitative analyses of paranasal sinuses and recesses successfully retrieved a taxonomic differentiation between the traditional species *D. kappleri*, *D. pilosus* and *D. novemcinctus* but failed to recover diagnostic features between the disputed and morphologically similar *D. septemcinctus* and *D. hybridus*. Most interestingly, the high variation detected in our large sample of *D. novemcinctus* showed a clear geographical patterning, with the recognition of three well-separated morphotypes: one ranging from North and Central America and parts of northern South America west of the Andes, one distributed across the Amazonian Basin and central South America, and one restricted to the Guiana Shield.

**Discussion.** The question as to whether these paranasal morphotypes may represent previously unrecognized species is to be evaluated through a thorough revision of the *Dasypus* species complex integrating molecular and morphological data. Remarkably, our recognition of a distinct morphotype in the Guiana Shield area is congruent with the recent discovery of a divergent mitogenomic lineage in French Guiana. The inflation of the second medialmost pair of caudal frontal sinuses constitutes an unexpected morphological diagnostic feature for this potentially distinct species. Our results demonstrate the benefits of studying overlooked internal morphological

structures in supposedly cryptic species revealed by molecular data. It also illustrates the under-exploited potential of the highly variable paranasal sinuses of armadillos for systematic studies.

## INTRODUCTION

Detection of cryptic diversity and pertinent delimitation of extant taxonomic entities constitute a major challenge of current-day biological research as it may have critical implications on biodiversity conservation policies (*Carstens et al., 2013*). Cryptic species can be defined as ''two or more species that are, or have been, classified as a single nominal species because they are at least superficially morphologically indistinguishable'' (*Bickford et al., 2007*: 149). According to this definition, the absence of diagnostic morphological characters may have impeded the recognition of species. Depending on the case, this absence might be real (i.e., populations of two cryptic species do not differ significantly in their entire anatomy) or spurious (i.e., morphological differences have been overlooked).

Advanced methods of micro computed tomography (μCT) now enable an unprecedented assessment of internal anatomical structures, which can help uncover previously concealed morphological differences between taxa. The development of these non- destructive methods permits internal anatomy to be easily and systematically investigated in many taxa. These methodological improvements offer great opportunities for morphology-based phylogenetic research. In mammals, internal cranial structures certainly present a great wealth of phylogenetically informative anatomical features (e.g., *Farke, 2010*; *Macrini, 2012*; *Ruf, 2014*; *Billet, Hautier & Lebrun, 2015*), possibly as many as the external surface of the skull. There is therefore a possibility that this large proportion of previously poorly explored morphological data contains undetected morphological differences between alleged cryptic taxa.

The pan-American nine-banded armadillo (*Dasypus novemcinctus*) presents the largest distribution of any living xenarthran species (*McDonough & Loughry, 2013*), and constitutes an interesting model for Neotropical phylogeography. Several subspecies (five to seven) have been recognized within this species but their delineation and recognition are not consensual (*Cabrera, 1958*; *McBee & Baker, 1982*; *Wetzel et al., 2008*; *McDonough & Loughry, 2013*). In fact, most potential diagnostic characters for these subspecific distinctions are seldom detailed, often inconstant, or based on a limited number of observations (e.g., *Peters, 1864*; *Allen, 1911*; *Lönnberg, 1913*; *Hamlett, 1939*; *Hooper, 1947*; *Russell, 1953*). Taxonomy and phylogeny in long-nosed armadillos (Dasypodidae, sensu *Gibb et al., 2016*) particularly suffer from strong disagreement between morphological and molecular data (*Castro et al., 2015*; *Gibb et al., 2016*). Even though it was more focused on higher taxonomic levels, a recent study suggested the existence of an unrecognized species in French Guiana based on mitochondrial data (incl. mitogenomes)

(*Gibb et al., 2016*). The Guianan entity has never been distinguished from other *D. novemcinctus* on a morphological basis, and might thus represent another striking case of cryptic species.

In order to explore if internal parts of the skull contain a useful phylogenetic signal, we investigated the internal paranasal sinuses and recesses (*Rossie, 2006*), whose complex structure has largely been ignored by morphologists working on the systematics of long-nosed armadillos (genus *Dasypus*). Based on µCT images of skulls, we reconstructed virtually the entire network of paranasal spaces in *Dasypus* species with a particular focus on specimens of *D. novemcinctus* covering the entire geographic range of the species. The observed patterns are described and discussed considering traditional taxonomic entities of long-nosed armadillos and in light of most recent molecular findings. A focus is made on the discriminatory power of these concealed characters in armadillos and on their utility for diagnosing taxonomic units previously regarded as cryptic.

## MATERIALS & METHODS

### Specimens and µCT-scanning

The total number of investigated specimens is composed of 136 skulls of *Dasypus* spp. harvested from various institutions worldwide (see details in Table S1), among which 112 were identified as *D. novemcinctus,* 1 as *D. sabanicola,* 13 as *D. kappleri*, 4 as *D. hybridus*, 3 as *D. septemcinctus*, and 3 as *D. pilosus.*

Among this sample, we virtually reconstructed the internal paranasal spaces in 51 µCT-scanned specimens belonging to *D. novemcinctus* ($n = 47$), *D. kappleri* ($n = 1$), *D. hybridus* ($n = 1$), *D. septemcinctus* ($n = 1$), and *D. pilosus* ($n = 1$). Among the 47 *D. novemcinctus* specimens, 7 were considered juveniles, including a potential stillborn (AMNH 33150); the 40 remaining being adults or subadults. Age classes (juveniles vs subadults or adults) were determined based on the stages of eruption of the teeth (*Ciancio, Castro & Asher, 2012*), on suture closure, and on size. The 47 *D. novemcinctus* specimens came from: United States ($n = 3$), Mexico ($n = 4$), Guatemala ($n = 1$), Nicaragua ($n = 1$), Costa Rica ($n = 1$), Panama ($n = 1$), Colombia ($n = 6$), Venezuela ($n = 2$), Ecuador ($n = 2$), Peru ($n = 2$), Bolivia ($n = 2$), Paraguay ($n = 1$), Guyana ($n = 3$), Suriname ($n = 2$), French Guiana ($n = 3$), Brazil ($n = 12$; see a list of different states in Table S1), and Uruguay ($n = 1$). Digital data of all 51 specimens were acquired using X-ray micro computed tomography (µCT). Most specimens were scanned on the X-ray tomography imagery platform at the Université de Montpellier (France) and on the µCT-scan platform of the Imaging and Analysis Centre of the British Museum of Natural History (London, UK); one (MNHN.ZM-MO 2001.1317) was scanned at the Museum National d'Histoire Naturelle (France) in Paris (AST-RX platform). Detailed information about the scans and acquisition parameters can be found in Table S1. Three-dimensional reconstructions and visualizations of the frontal sinuses were performed using stacks of digital µCT images with AVIZO v. 6.1.1 software (Visualization Sciences Group, Burlington, MA, USA).

An additional subset of 65 *D. novemcinctus*, 1 *D. sabanicola*, 12 *D. kappleri*, 3 *D. hybridus*, 2 *D. septemcinctus* and 2 *D. pilosus* specimens was added to the sample mentioned above. These additional specimens correspond to:

(i) skulls not available for µCT-scanning but that allowed observing frontal sinuses boundaries through bone transparency through direct observations or photographs (NB: this was not possible for all observed skulls, some being insufficiently prepared or having no transparency of the frontal bone);

(ii) µCT-scanned skulls whose paranasal cavities were not virtually reconstructed but their boundaries observed with ISE-Meshtools (*Lebrun, 2014*), with an artificial cutting of the specimen following a coronal section and with the software Landmark 3.6 (available at http://graphics.idav.ucdavis.edu/research/EvoMorph; Institute for Data Analysis and Visualisation, Davis, CA, USA) with the option transparent surface rendering.

These two methods helped us to increase the number of investigated specimens and, most particularly, to include a paratype specimen of *D. sabanicola* (*Mondolfi, 1968*) (Table S1) (see discussion for a word on the status of *D. sabanicola*). In order to identify a cavity observed with these alternative methods, we used similarities in position, shape, and topographical relationships with sinuses or recesses defined in virtually reconstructed specimens.

## Nomenclature of paranasal anatomy

To our knowledge, no detailed description of paranasal cavities exists for extant armadillos, except for histological slices in *Reinbach (1952a)* and *Reinbach (1952b)*. A maxillary recess is mentioned and figured in *Euphractus* (*Wible & Gaudin, 2004*) and brief notes were reported on the soft paranasal anatomy in *Dasypus* (*Soares da Silva et al., 2016*). The most extensive work on paranasal spaces in Cingulata concerns in fact some glyptodonts (*Fernicola et al., 2012*), whose sinuses are very different from that of long-nosed armadillos. For these reasons, our nomenclature follows several conventions used in other taxa, as detailed below. The standard practice for paranasal sinuses is to name them after the bones they excavate (*Novacek, 1993*); we respected this practice for all the cavities we detected (i.e., both sinuses and recesses). The identity of the bones housing these cavities was determined through the examination of juvenile specimens that display clearly visible bone sutures. Following recent works by *Maier (2000)*, *Rossie (2006)*, and *Farke (2010)*, we made a distinction between sinus and recess for paranasal cavities. Paranasal sinuses are pneumatic and mucosa-lined spaces that are located in the bones surrounding the nasal chamber (*Rossie, 2006*; *Curtis & Valkenburgh, 2014*). Contrary to sinuses that are found between two layers of cortical bones (e.g., frontal), paranasal recesses are defined as simple concavities of the nasal cavity, and are not associated with active bone removal (*Farke, 2010*; *Rossie, 2006*). Hereafter, we employed the terms sinus or recess accordingly. However, some cavities may apply to both definitions, with the posterior part expanding into the bone while the anterior part only represents a concavity. In order to avoid confusion in giving two names to a single structure, we called sinuses the cavities that were at least partly comprised between two layers of a cortical bone. Because there are often several sinuses or recesses within a given bone (e.g., frontal, maxillary), we also used the English equivalents of positional terms of the Nomina Anatomica Veterinaria (*NAV, 2005*) for the paranasal sinuses when feasible (e.g., caudal frontal sinus). It was not possible to elaborate robust homology hypotheses for all cavities of the paranasal region because they are found in large

numbers and may represent neoformations when compared to the common terminology. Consequently, we complemented the common terminology with a numbering system that allows distinguishing the numerous frontal recesses and sinuses found in long-nosed armadillos. The terminology for turbinal bones, which are only briefly mentioned for spatial localization of paranasal cavities, is based on *Van Valkenburgh, Smith & Craven (2014)* and *Maier & Ruf (2014)*.

## RESULTS

### Observations common to all long-nosed armadillos (genus Dasypus)

In all investigated long-nosed armadillos, paranasal sinuses and recesses consistently excavate the same three bones of the cranial face and vault: the lacrimal, the maxillary and the frontal. Sinuses are present only in the frontal bone of *D. novemcinctus*, *D. pilosus*, and *D. kappleri*; they are absent or weakly marked in *D. hybridus* and *D. septemcinctus* (Fig. 1). Only the posterior pneumatic parts of the frontal bone form sinuses whereas, more anteriorly, the pneumatization of the frontal bone forms recesses, which are in direct contact with the underlying turbinals all along their anteroposterior length. In most adult individuals of *D. novemcinctus* (see more details below) and *D. kappleri*, the frontal sinuses are almost entirely bordered by the posterior part of the fronto- and ethmoturbinals ventrally. In all species, the frontal sinuses and recesses regularly increase in height toward the front. The number of sinuses and recesses varies intragenerically and these structures will be described hereafter.

   In all *Dasypus* species, recesses are positioned dorsolaterally in the paranasal cavity. These recesses represent large free-of-bone spaces in the nasal cavity, generally separating turbinal bones medioventrally from the bones that build up the cranial walls. The lacrimal recesses are in contact with the mass of fronto- and ethmoturbinals medially, whereas the maxillary recesses are bordered by the naso- and maxilloturbinalsventromedially . Two distinct lacrimal recesses are invariably present and are separated by the bony cover of the nasolacrimal duct. Two recesses, variably individualized, excavate the maxillary bone (Fig. 1) and are bordered by the nasolacrimal duct ventrolaterally. These maxillary cavities may well be homologous to the maxillary sinus found in other mammals, but they are here designated as recesses following the rationale specified in the 'Material and Methods'.

   In addition to the taxonomic and geographic variation described below, variable levels of intra-individual asymmetry appear to affect all species and all paired paranasal spaces under consideration here. This asymmetry is not directional, it is ubiquitous and present in most if not all specimens, which suggests a case of fluctuating asymmetry (*Van Valen, 1962*). The species *D. kappleri* seems to be characterized by stronger levels of asymmetry than the species *D. novemcinctus* (see below).

### Juveniles of *D. novemcinctus*

In addition to delivering critical information on the identity of bones housing the various sinuses and recesses, the study of juvenile individuals provided some clues on the growth pattern of the paranasal pneumatization in the nine-banded armadillo. Juveniles show a less tight medial contact between paired medial sinuses, such as the rostral frontal recesses

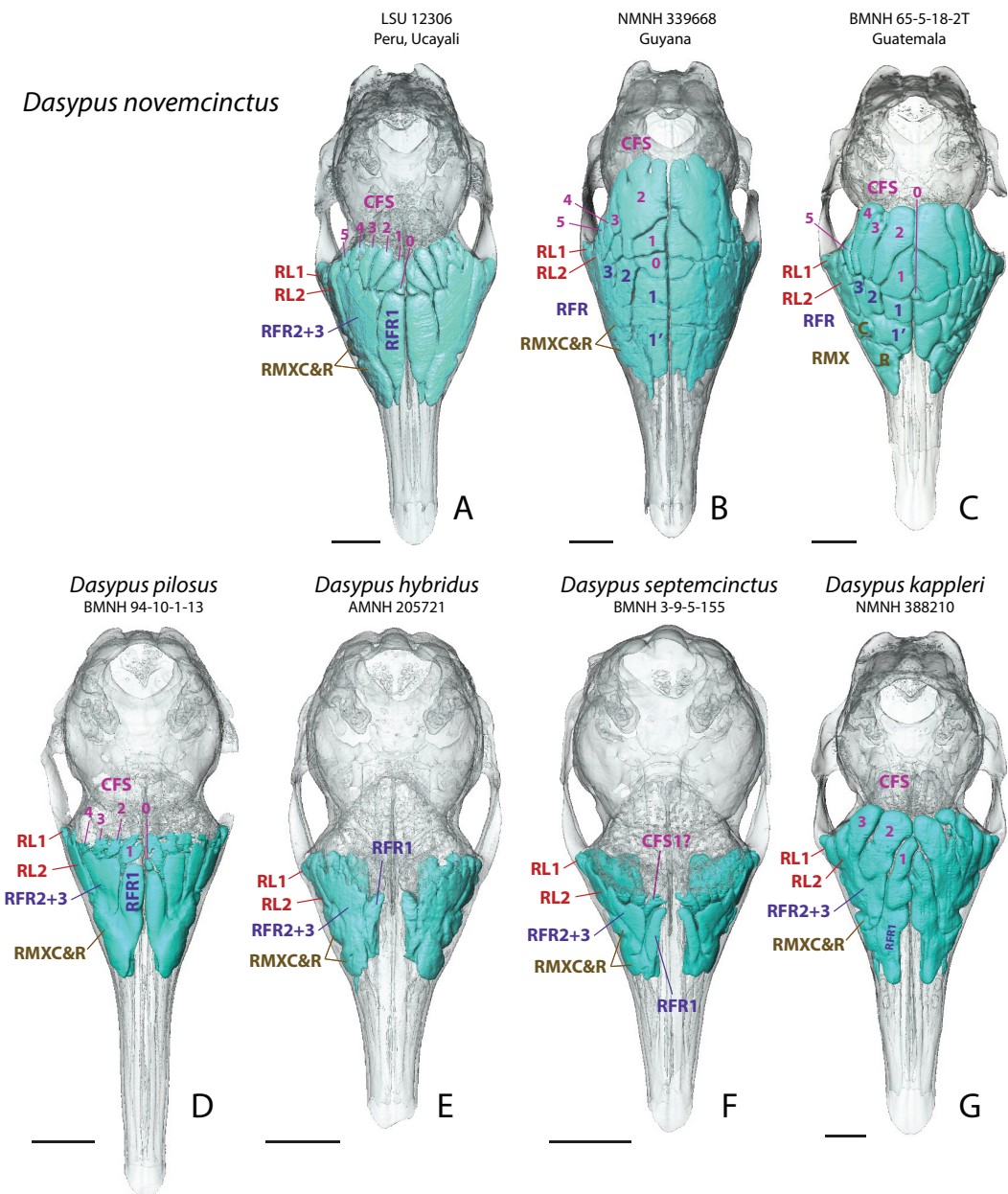

**Figure 1   Dorsal views of virtually reconstructed Dasypus skulls with internal paranasal sinuses.** Dorsal views of virtually reconstructed skulls of long-nosed armadillos species, with bone transparency showing internal paranasal sinuses and recesses in light blue. (A–C), *D. novemcinctus*; (D) *D. pilosus*; (E) *D. hybridus*; (F) *D. septemcinctus*; (G) *D. kappleri*. See a list of anatomical abbreviations at the end of the article. Scale-bar: 10 mm.

(which may split in RFR1 and RFR1′, see below) or the caudal frontal sinuses (Fig. 2). Compared to adults, caudal frontal sinuses (CFS) are less expanded posteriorly in juveniles and do not lie above the most posterior part of the mass of fronto- and ethmoturbinals.

A very young specimen (likely a stillborn, AMNH 33150) shows that very early ontogenetic phases of paranasal pneumatization start with a weak individualisation of

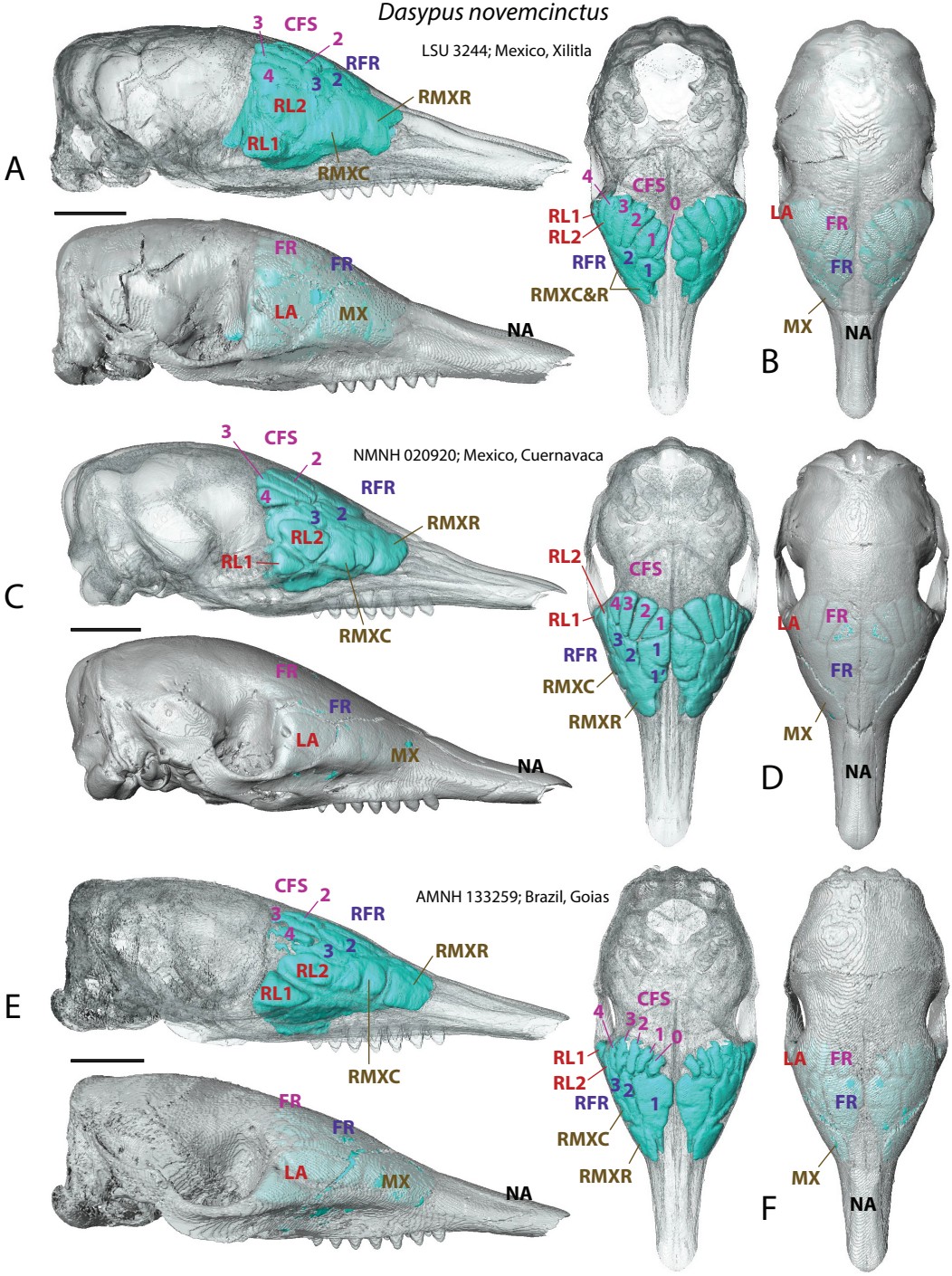

**Figure 2  Paranasal sinuses and recesses in juvenile individuals of *Dasypus novemcinctus*, virtual reconstructions of skulls in lateral (A, C, E) and dorsal views (B, D, F), with and without bone transparency.** See a list of anatomical abbreviations at the end of the article. Scale-bar: 10 mm.

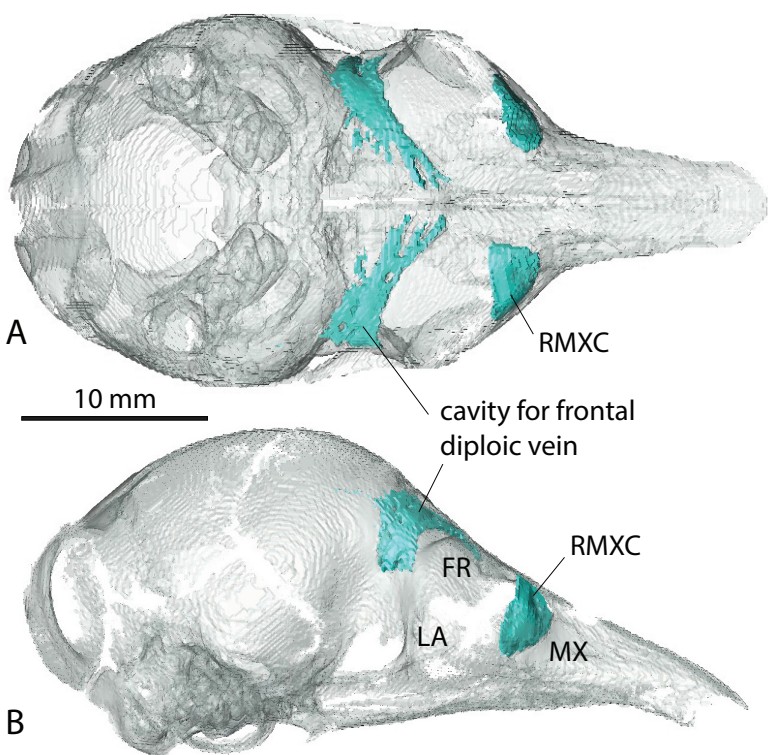

**Figure 3 Virtual reconstruction of the skull of the stillborn specimen AMNH 33150, *Dasypus novemcinctus*, with bone transparency leaving the caudal maxillary recess and cavity for the frontal diploic vein apparent.** (A) dorsal view; (B) lateral view.

the caudal maxillary recess, whereas no other sinus or recess is individualized and the turbinals are not yet ossified. The large cavity excavated in the posterior part of the frontals in this specimen does not represent a sinus but a transverse canal, presumably for the frontal diploic vein (*Wible & Gaudin, 2004*) (Fig. 3). Other juveniles in our dataset clearly correspond to later ontogenetic stages, as indicated by their size: LTC= AMNH33150 38,93 mm; LSU3244 66,14 mm; NMNH 020920 72,89 mm; AMNH133259 68,85 mm (NB: LTC ∼90–105 mm in adults; L Hautier, 2017, unpublished data). This age difference is confirmed by their stage of dental eruption: first decidual bicuspid tooth erupting in AMNH 33150; dP1-dP7 present in LSU3244 and NMNH 20920; dP1-dP7 and alveolus of M1 present in AMNH 133259 (see (*Ciancio, Castro & Asher, 2012*). The juvenile series shows that paranasal pneumatization and turbinal ossification just barely started in perinatal stages lesser than 40% adult skull length (AMNH 33150; Fig. 3) whereas these structures are well-developed in later stages with dp1-dp7 erupted and with ∼70% of adult skull length (Fig. 2).

## Observations common to all adults of nine-banded armadillos (*D. novemcinctus*)

Skulls of adult *D. novemcinctus* are more pneumatized than juvenile ones (Figs. 2 –4). All adult *D. novemcinctus* present a similar pattern of sinuses: posteriorly, a number of 5–6 paired CFS generally cover dorsally the posterior part of the mass of fronto- and

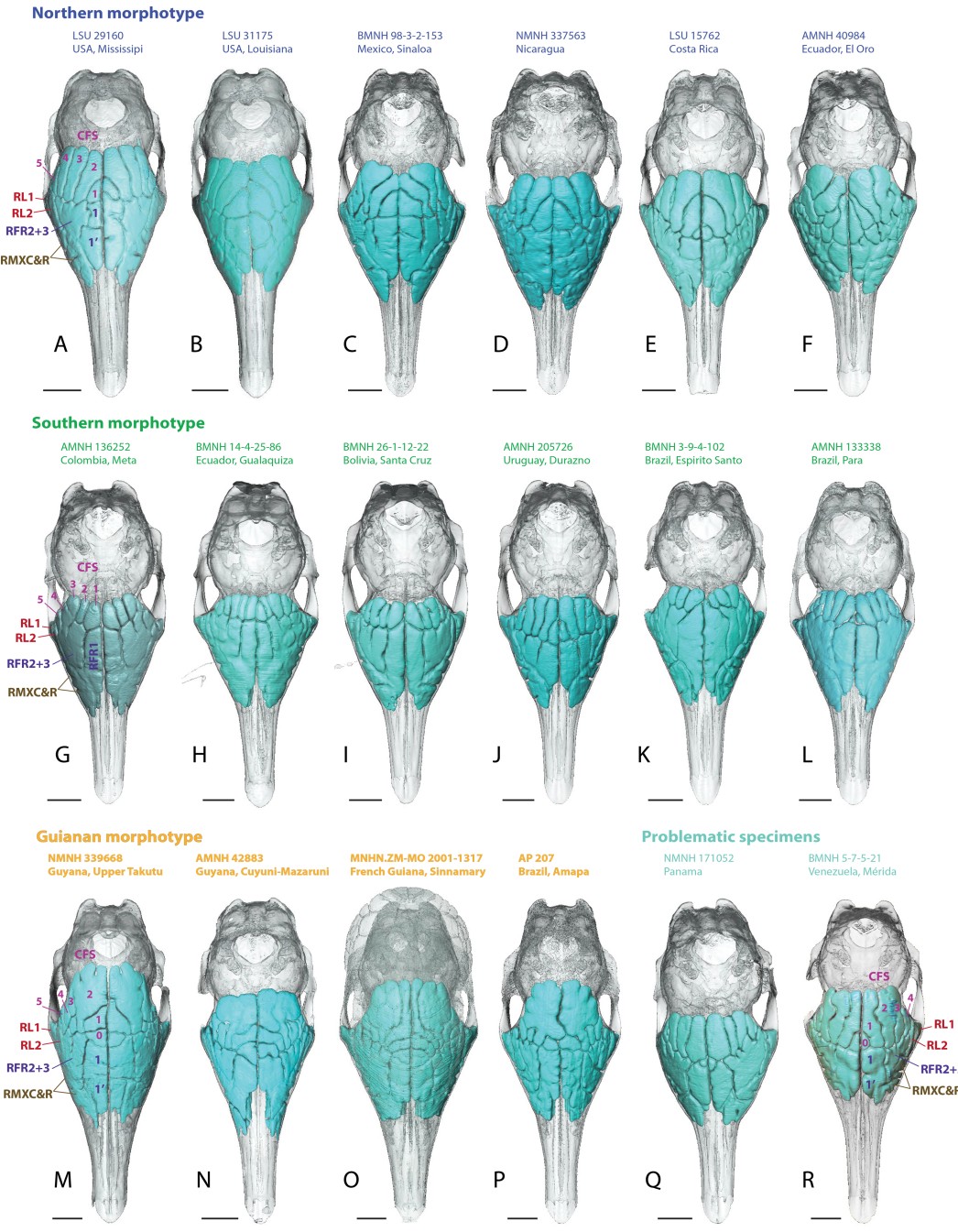

**Figure 4** **Dorsal views of virtually reconstructed skulls of adult specimens of *Dasypus novemcinctus* clustered by morphotypes of paranasal anatomy as described in the text.** (A–F), northern morphotype; (G–L), southern morphotype; (M–P), Guianan morphotype; (Q–R), problematic specimens (see text). Bone transparency leaves apparent the paranasal recesses and sinuses in light blue. See a list of anatomical abbreviations at the end of the article. Scale bar: 10 mm.

ethmoturbinals (Figs. 1, 4 and 5). One medialmost and often reduced pair of CFS was identified as variably present and designated as CFS 0, in order to start numbering from 1 for the invariably present CFS. The set of CFS form a continuous transversal chain of dorsal paranasal spaces between the orbits (Fig. 5). While the posterior part of each CFS is comprised between two layers of cortical bone, the anterior part is always bordered ventrally by the fronto- and ethmoturbinals (Fig. 5). Anterior to the CFS, the frontal bone houses several pairs of rostral frontal recesses (RFR). These recesses show variable shapes (see below), but can be at least divided in two main areas: a medial recess generally elongated (RFR1) and/or subdivided anteroposteriorly (RFR1 & RFR1′), and a recess or group of recesses that excavate the frontal bone more laterally up to the lacrimal recesses (RFR2-3) (Fig. 1). Topographical criterions were used for establishing homologies and numbering of the CFS, with the CFS 0-1 always located directly posterior to the RFR1 and the CFS2-4 located posterior to the RFR2-3. In our sample of *D. novemcinctus*, the RFR1 are always bordered posteriorly by one or two pairs of CFS. When two pairs are present, the medialmost (or anteriormost in a few specimens; see below), CFS 0, is always the smallest. Therefore, we consider that the medialmost pair is CFS1 when only one pair is present (Figs. 1 and 4).

## Northern morphotype of *D. novemcinctus*

Thirty nine (39) specimens from North and Central America and from the Pacific coast of eastern Ecuador are attributed to this morphotype (Figs. 1, 4 and 6). Specimens attributed to this group originate from (in alphabetical order of countries): Belize, Colombia (Antioquia Department), Costa Rica, Ecuador (Provinces El Oro and Pichincha), Guatemala, Honduras, Mexico (Sinaloa, Tabasco, Oaxaca, Colima, Jalisco, and an undetermined locality (NMNH179172) for the adults, San Luis Potosi and Morelos for the juveniles), Nicaragua, and USA (states of Mississipi, Texas, Florida, Kansas, and Louisiana). The main diagnostic feature for this group is the anteroposterior elongation of the CFS2 to 5; in addition, the left and right CFS2 are obliquely orientated and contact each other posterior to the CFS1. Another distinctive feature of this morphotype is the subdivision and the relative shortening of RFR1. As for the Southern morphotype (see below), the number of CFS pairs in this group varies from 5 to 6, because the CFS0 pair is either very reduced or absent. The CFS1 are rather small, shorter than the more lateral CFSs and bordered posteriorly by the contacting pair of CFS2. The CFS2 and/or CFS3 are the largest CFSs within this group, and though they do not contact posteromedially, each CFS3, similarly to the CFS2, bends or orientates obliquely toward the midline posteriorly. The CFS4 can be as elongated as the CFS2-3 or slightly smaller; the CFS5 are more reduced. The median rostral frontal recesses are subdivided into two anteroposterior pairs RFR1 and RFR1′ and contrast with the long RFR1 of the Southern morphotype. The posterior pair, the RFR1, apparently forms earlier within the ontogenetic sequence as it is clearly more developed than RFR1′ in two juvenile specimens from Mexico (Fig. 2). The shape of RFR1 in adults is rather square whereas the anterior pair, RFR1′, is usually slightly more elongated anteroposteriorly. Lateral to the RFR1 and RFR1′, the RFR2 and RFR3 are often well separated (distinction often better marked than in the Southern group); the RFR3

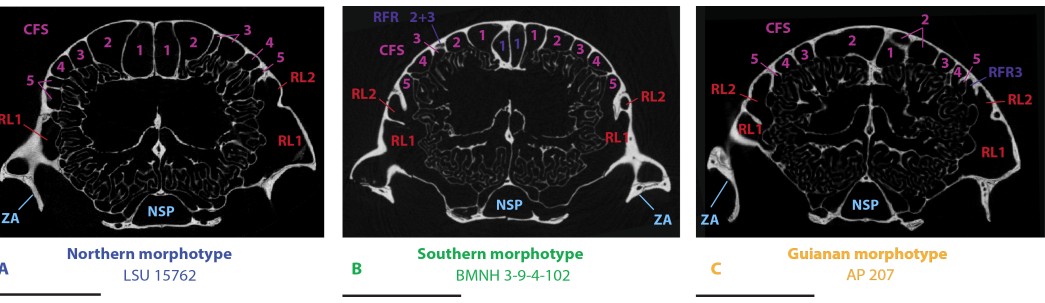

**Figure 5** μCT transversal slices through the skull of *D. novemcinctus* individuals showing details of the internal paranasal anatomy for each morphotype (A, northern; B, southern; C, Guianan). Slices were made at similar transversal locations at the posterior end of the anterior root of the zygomatic arch. See a list of anatomical abbreviations at the end of the article. Scale-bar: 10 mm.

is immediately lateral to the RFR2. The same applies to the caudal and rostral maxillary recesses (RMXC & RMXR), which are often better separated in this morphotype compared to the Southern morphotype; the caudal maxillary recess is posterolateral to the rostral one and located just anterior to the lacrimal recess 2.

The specimen AMNH 40984 from southwest Ecuador is attributed to this Northern group because of the presence of large and posteriorly convergent CFS2 and short and subdivided RFR1-RFR1′ (Fig. 4). The other specimen from western Ecuador, BMNH 16.7.12.37, also shows these characters through bone transparency. Nevertheless, on the virtual reconstruction of AMNH 40984, the lateral RFR2-3 appear more subdivided than in other members of this morphotype, and also more than in other morphotypes. In addition, the anterior edge of its medialmost CFS (CFS 0) is shifted anteriorly. These unique characters could not be checked on the specimen observed on photos only.

## Southern morphotype of *D. novemcinctus*

Fifty one (51) specimens (incl. the paratype of *D. sabanicola*) spanning the Amazon Basin (excluding the Guiana Shield) and including the southernmost distribution of the species in Uruguay show a distinct pattern of paranasal spaces (Figs. 4 and 6). Specimens attributed to this group thus span the Southern range of *D. novemcinctus* and originate from the following countries (in alphabetical order): Bolivia (states of Beni, Pando and Santa Cruz), Brazil (states of Amazonas, Para, Goias, Santa Catarina, Mato Grosso, Mato Grosso do Sul, Minas Gerais, São Paulo, Rio Grande do Sul and Espirito Santo), Colombia (Meta and Magdalena departments), Ecuador (Morona Santiago Province), Paraguay, Peru (regions of Ucayali, Ayacucho and San Martín), Uruguay, and Venezuela (states of Anzoátegui and Apure (state of the paratype of *D. sabanicola*)). The distinctive features of the Southern pattern of paranasal spaces mostly consist in an anteroposteriorly reduced posterior chain of caudal frontal sinuses and an elongated rostral frontal recess 1 (Figs. 1 and 4). In this group, the CFS0 are variably present. When present, they are generally smaller than the more lateral CFS1 to 5. The CFS1 always contact each other medially, or just at their posteromedial corner if the CFS0 are present. In most specimens, the CFS are much shorter anteroposteriorly than the rostral frontal recess RFR1, usually around a 1/2 or 1/3 ratio. Some specimens referred to this morphotype show more balanced ratios but the

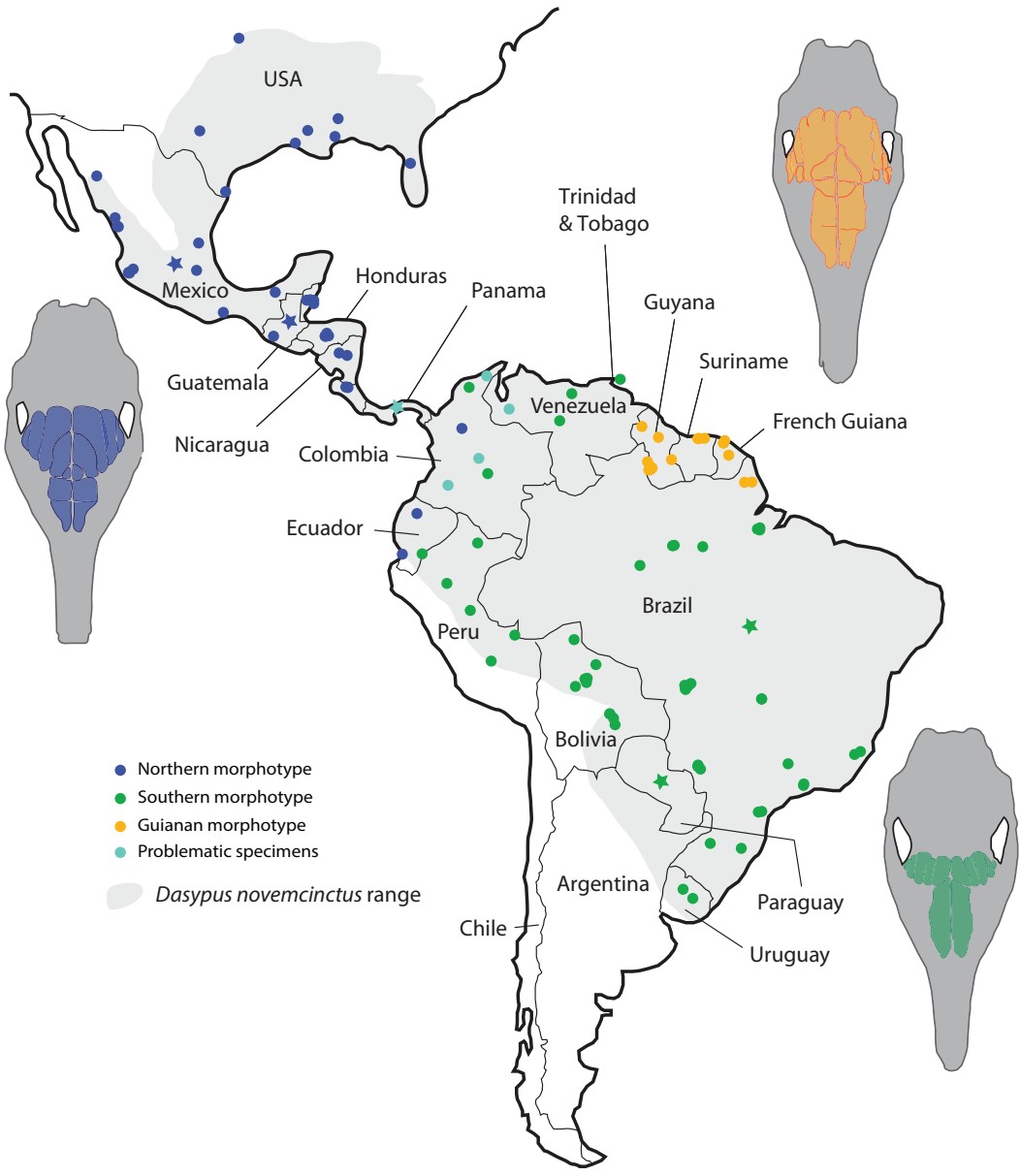

**Figure 6** **Summary map showing the geographical distribution of nine-banded armadillo specimens (*Dasypus novemcinctus*) investigated in this study and their attribution to a paranasal morphotype.** Each morphotype is represented by a schematic dorsal view of skulls (in grey) on which the frontal paranasal sinuses and recesses are drawn (in blue, yellow, or green for each morphotype). Specimens reported with a star denote the absence of geographical information besides the country of origin.

anteroposterior length of the CFS never exceeds that of the RFR1. The CFS1 to 4 are usually of similar length and width, the areas of the CFS 1 and 2 may just slightly exceed that of the others in average. The lateralmost caudal frontal sinus that lies immediately medial to the orbital rim, i.e., the CFS5, is generally shorter than the other CFS.

Additionally, many individuals of that group show a weak distinction or even a fusion between the RFR2 and RFR3 (Figs. 1 and 4). Though the RFR2 and RFR3 are in average

less separated than in other groups of *D. novemcinctus*, this character is not stable within the Southern group. Ontogenetic data are unfortunately lacking to state on a possible influence of development on this feature. When not completely fused, the RFR 2 and 3 are often separated by a short bony ridge posteriorly.

The maxillary recesses are located dorsal to the nasolacrimal duct and (antero)lateral to the RFR1. As for the RFR2 and RFR3, the caudal and rostral maxillary recesses are variably distinct; in average, they are less separated than in the other groups.

## Guianan morphotype of *D. novemcinctus*

Seventeen (17) specimens are attributed to this group and originate from: Brazil (Amapa State), French Guiana, Guyana and Suriname. The most conspicuous diagnostic feature of this group is the strong inflation of the CFS2, which is by far the largest caudal frontal sinus (Figs. 1, 4–6). These hypertrophied CFS2 occupy most or all the pneumatized frontal area between the orbits up to the level of the anterior edge of the posterior zygomatic root posteriorly. More precisely, it is the posterior part of the CFS2 that is hypertrophied and borders all or most other CFSs on their caudal side. The anterior part of the CFS2, which is sandwiched between CFS1 and CFS3, is as narrow as the CFS3-4. The outline of CFS2 often exhibits a complex irregular pattern and pairs are clearly asymmetric in all specimens (Fig. 4).

The CFS0 are variably present as in other groups. When present, they are rather small, clearly elongated transversally and entirely bordered by the CFS1 posteriorly. In comparison to other morphotypes, the CFS0 are slightly shifted anteriorly relative to other CFSs. The size and shape of the CFS1 is also variable. In specimens with no CFS0, the CFS1 are much reduced and rather square-shaped (AP 207; Fig. 4). In specimens with CFS0, the CFS1 are relatively larger and can extend as far posteriorly as the hypertrophied CFS2. In such case (MNHN.ZM-MO 2001-1317), they are transversally thin and sandwiched between the pairs of CFS2 (Fig. 4). In one specimen (MNHN.ZM-MO 1996-587), the CFS1 seems to be partly fused posteromedially with the CFS2 on both sides. The CFS3 and 4 are considerably smaller than the CFS2, and are similar in size as in the Southern morphotype. The CFS5 is generally smaller than CFS3-4, and can be absent (e.g., MNHN.ZM-MO 2001-1317).

The configuration of the RFR1 is variable, as they seem to be irregularly divided into an anterior RFR1′ and posterior RFR1. The limits between these two subdivisions are not only variable between individuals of this morphotype, they are also labile intra-individually: the subdivisions may be marked on one side of the skull, not on the other (ROM 32275) or it may follow another path (MNHN.ZM-MO 1995-553), or the boundaries may be irregularly marked overall (marked on some portion, then absent, and then marked again a little farther away; NMNH 339668). As in the Southern morphotype, the RFR2 and 3 are poorly distinguishable in most Guianan specimens (but this is also variable), except for a posterior demarcation that is almost always present.

The lacrimal recesses 1 and 2 are very similar to that of the other morphotypes, also delimited by the nasolacrimal duct. The intensity of the separation between the rostral and caudal maxillary recesses is variable, but these recesses are otherwise very similar to other morphotypes.

## Problematic specimens from Panama, Venezuela and Colombia

Five specimens (AMNH 32356, 37356, NMNH 281290 from Colombia, BMNH 5-7-521 from Merida, Venezuela and NMNH 171052 from Panama) show somewhat intermediate morphologies between the Southern and Northern morphotypes (Fig. 4). They all present CFS and RFR1 of similar length and do not show a medially contacting pair of CFS2 (though close in AMNH 37356). The RFR1 are usually not subdivided, except in the Venezuelan specimen and, to a lesser extent, in the Colombian AMNH 32356. The CFS2 and 3 do not represent the largest CFS except in the Colombian AMNH 32356 and 37356. These specimens therefore show a combination of features characterizing the Southern and Northern groups.

In addition, the probable stillborn specimen AMNH 33150 from Colombia could not be referred to any morphotype because its paranasal spaces are not fully developed yet (Fig. 3).

## Other *Dasypus* species
### Greater long-nosed armadillo (D. kappleri)

The *D. kappleri* specimens present a large pneumatization of their paranasal region, as in *D. novemcinctus*. However, all *D. kappleri* specimens exhibit less numerous, but wider and longer finger-shaped CFS than *D. novemcinctus*. In fact, this may be due to various partial or complete fusions of the CFS with the RFR, which we tentatively identify as follows: the CFS1 and RFR1 are fused and occupy the medialmost region (but not posteriorly in some specimens; see below), the CFS2-3 are fused with the RFR2-3 (Fig. 1). We consider that the caudal end of these structures are homologous to the CFS since it is found between two layers of frontal bone. In some specimens, a blunt bony bridge still marks a separation between these sinuses and recesses. This general pattern is typical of the species, yet the arrangement of paranasal cavities largely varies intraspecifically. Two groups can be distinguished: specimens from the Guiana Shield display fused CFS1-RFR1 that reach the posterior boundary of the other CFS, whereas specimens from more western locations have the right and left CFS2 that contact posteriorly in the midline (Fig. S1). The relative sizes of the CFS also vary a lot, but the fused CFS2-3 with RFR2-3 are in most cases the largest ones. In addition, most of these recesses show a substantial amount of asymmetry, probably higher than in *D. novemcinctus*. Other recesses are grossly similar in size and location to those described for *D. novemcinctus*.

### Hairy long-nosed armadillo (D. pilosus)

The investigated specimens of *D. pilosus* probably resemble the most the *D. novemcinctus* groups (Fig. 1). Conversely to *D. kappleri*, specimens of *D. pilosus* show caudal frontal sinuses well-individualized from the rostral frontal recesses. However, there is some variation in the shape of the caudal frontal sinuses. In the only scanned specimen of *D. pilosus*, the caudal frontal sinuses are barely recognizable as they do not excavate the frontal bone (not found between two layers of frontal bone); they are located just dorsal to the mass of fronto- and ethmoturbinals. In fact, they rather represent thin cell-shaped recesses with irregular outlines, which are sandwiched between the frontal bone dorsally and the fronto- and ethmoturbinals ventrally. Conversely, the two additional specimens of *D. pilosus* observed through bone transparency show slightly longer caudal frontal

sinuses/recesses that are better delineated. In any case, these caudal cell-shaped frontal sinuses/recesses are in all three specimens comparable to the CFS of *D. novemcinctus* groups in their number (4 to 5 pairs), dorsal outline and location. The rather short anteroposterior extent and reduced mediolateral width of these CFS-like structures in *D. pilosus* are most reminiscent of the pattern seen in the Southern group of *D. novemcinctus*. The more anterior recesses resemble the RFR1 and RFR2-3 of the same Southern group, especially the un(sub)divided and elongated RFR1. Remarkably, the lacrimal recesses are more elongated anteroposteriorly than in other *Dasypus* species.

### Southern long-nosed armadillo (D. hybridus) and seven banded armadillo (D. septemcinctus)

These two species are here described together because they exhibit strong similarities in their pattern of paranasal cavities and could not be distinguished in our sample. These two small-sized species show the least pneumatized skulls among our adult sample of *Dasypus*. Both species present RFR1 that are transversely narrow and curved, never in contact medially, and thus partly recall the configuration seen in young *D. novemcinctus* specimens (see above) (Figs. 1 and 2). In addition, specimens of both species have poorly defined CFS, i.e., the fronto- and ethmoturbinals fill in most of the space just ventral to the cranial vault made by the frontals and the CFS are very thin dorsoventrally. The frontal bones are in fact poorly pneumatized and show a thin diploe. Other cavities (rostral frontal recesses, lacrimal and maxillary recesses) show a pattern and extent grossly similar to that of other *Dasypus* species.

## DISCUSSION

### Distribution and significance of paranasal pneumatization in mammals and armadillos

*Paulli (1900a)* and *Paulli (1900b)* first provided detailed descriptions of paranasal cavities based on sagittal and transverse osteological sections of mammalian skulls. With the recent development of μCT and virtual modeling of internal structures, the paranasal sinuses and recesses could be more systematically and precisely studied in extant mammals such as vombatiform marsupials, carnivorans, artiodactyls, and primates (e.g., *Rossie, 2008*; *Farke, 2010*; *Curtis & Valkenburgh, 2014*; *Maier & Ruf, 2014*; *Sharp, 2016*). Though not yet thoroughly investigated with modern techniques, these structures are also known to occur in many other groups of placental mammals (*Paulli, 1900a*; *Paulli, 1900b*; *Edinger, 1950*; *Novacek, 1993*) and may constitute convergently lost symplesiomorphic placental features (*Foster & Shapiro, 2016*).

The ubiquitous distribution of these structures in several clades of amniotes (*Witmer, 1999*) long raised questions regarding the potential functional role of paranasal pneumatization. As noted by *Farke* (*2010*: 988), cranial pneumatization such as paranasal sinuses "remains one of the most functionally enigmatic and debated structures within the vertebrate skull". Indeed, researchers have long speculated on the potential functional role of these air-filled chambers, and proposed a wealth of hypotheses (*Blanton & Biggs, 1968*; *Blaney, 1990*; *Marquez, 2008*), most of which remain, as of today, untested. However, one

of the current dominating hypotheses regards sinuses as functionless structures influenced by constraints inherent to bone growth and patterning (*Witmer, 1997*; *Smith et al., 2005*; *Farke, 2010* and citations therein). In fact, sinuses may just opportunistically fill space where bone is not mechanically necessary (*Curtis & Van Valkenburgh, 2014*) and reduce skull mass in return (*Curtis et al., 2015*). This might be compatible with the fact that the presence and extent of sinuses may, at least in some instances, be linked to size increase and to the shape of the bone in which they are contained (*Weidenreich, 1941*; *Zollikofer et al., 2008*; *Farke, 2010*; *Curtis et al., 2015*; *Krentzel & Angielczyk, 2016*; *Ito & Nishimura, 2016*; *Sharp & Rich, 2016*). Though these alternative architectural explanations do not preclude the existence of functional advantages (e.g., to dissipate mechanical stress during biting; *Tanner et al., 2008*), it seems that there is no overarching explanation for the function of sinuses (*Curtis et al., 2015*).

A substantial variation of paranasal sinuses shape and outline has long been noted in many taxa at the interspecific, intraspecific, and intra-individual levels (e.g., *Paulli, 1900a*; *Paulli, 1900b*; *Novacek, 1993*; *Farke, 2010*; *Curtis & Van Valkenburgh, 2014*). These observations clearly suggest that these structures have a non-negligible propensity to vary greatly in mammals. It is questionable whether or not their high variability (sensu *Hallgrímsson & Hall, 2005*) could make paranasal sinuses good markers of phylogenetic history. Interestingly, the highly variable shape and size of the frontal sinus in modern humans proved to be largely inherited from parents to children (*Szilvássy, 1982*) and is used in forensic science for individual and population identifications (e.g., *Kim et al., 2013*). At higher taxonomic levels, a significant phylogenetic signal was detected in the pattern of paranasal sinuses of primates and bovid artiodactyls (*Rossie, 2008*; *Farke, 2010*), but the size and shape of frontal sinuses were rather weakly linked with phylogenetic groupings in Carnivora (*Curtis & Van Valkenburgh, 2014*). Similarly, the diversity of maxillary sinuses in macaques was not linked to phylogeny (*Ito & Nishimura, 2016*) even though these structures were at least in part controlled by intrinsic genetic factors (*Ito et al., 2015*).

In the case of long-nosed armadillos, the clear discrete differences in patterns of paranasal sinuses observed between the different species and subgroups of *Dasypus* (*D. novemcinctus*, *D. kappleri*, *D. pilosus*, *D. septemcinctus* and *D. hybridus*) argue for a high discriminatory power and a good phylogenetic signal carried by these structures within the genus. The fluctuating asymmetry (*Van Valen, 1962*) tentatively identified for these structures in armadillos suggests that they are also impacted by random perturbations of developmental processes (*Klingenberg, 2010*). Curiously, early anatomical accounts of paranasal anatomy disagreed on the presence of sinuses in long-nosed armadillos. While *Cuvier (1845)* and *Weinert (1925)* correctly observed the presence of such structures in long-nosed armadillos, other early authors overlooked it (*Paulli, 1900b*; *Zuckerkandl, 1887* (as cited in *Weinert, 1925*)). In fact, frontal sinuses were still considered absent in armadillos as a whole in recent anatomical works (*Novacek, 1993*). Our results clearly contradict these considerations and investigation of paranasal cavities in some Chlamyphoridae, the sister group of Dasypodidae within Cingulata (*Gibb et al., 2016*), even reveals homoplastic evolution of these structures in armadillos. Frontal sinus or recesses are absent in the

extant chlamyphorid *Euphractus sexcinctus* (*Wible & Gaudin, 2004*) and some μCT-scanned specimens of *Cabassous unicinctus* (MNHN.ZM.MO 1953-457) and *Zaedyus pichiy* (MNHN.ZM.MO 1917-135) do not show any free-of-bone space between the frontal and the fronto- and ethmoturbinals (G. Billet, pers. obs., 2017). On the other hand, an extensive system of paranasal sinuses exists in the extinct glyptodont *Neosclerocalyptus* (*Fernicola et al., 2012*). Further comparisons are needed in extant and fossil forms (see sinuses in the fossil *D. punctatus*, *Castro et al., 2013*), as these structures might provide potentially interesting characters for the understanding of higher-level relationships within the order (*Delsuc et al., 2016*).

### Relevance of paranasal sinuses for the systematics of long-nosed armadillos

Our detailed investigation of paranasal cavities in *Dasypus* species revealed an important variation at different levels. We briefly described the ontogenetic pattern of the paranasal sinuses and recesses, which may start individualizing in perinatal stages (see also (*Reinbach, 1952a*; *Reinbach, 1952b*). Postnatal juvenile specimens show CFS that are less developed posteriorly when compared to adult specimens, revealing the late posterior growth of these structures. Second, as indicated above, adults show clear differences between traditionally recognized species, mostly in the configuration of the CFS and RFR. Besides the large variation seen within *D. novemcinctus* (see below), clear differences can be observed between *D. kappleri*, *D. pilosus* and the sister species *D. hybridus—D. septemcinctus*. The greater long-nosed armadillo (*D. kappleri*) probably has the most divergent morphology regarding these sinuses and recesses with the fusion of its CFS and RFR. In contrast, these structures are better separated in all other long-nosed armadillos reconstructed here. This is congruent with the early diverging position of *D. kappleri* in the phylogeny of long-nosed armadillos (*Gibb et al., 2016*). Our sample for *D. kappleri* is also characterized by a substantial variation, which is partly structured geographically: specimens from the Guiana Shield show a CFS1-RFR1 that reaches the posterior level of other CFS, whereas this is not the case in other specimens originating from more western areas in South America (Fig. S1). Interestingly, these two allopatric groupings are congruent with the new taxonomic subdivision proposed by *Feijo & Cordeiro-Estrela (2016)*, with a revised *D. kappleri* species restricted to the Guiana Shield area, and a new species (*D. pastasae*) found from the eastern Andes of Peru, Ecuador, Colombia, and Venezuela south of the Orinoco River into the western Brazilian Amazon Basin. These preliminary results now require a larger sample, including specimens referred to *D. beniensis* (*Feijo & Cordeiro-Estrela, 2016*), in order to further test species delimitation in the *D. kappleri* complex.

The pattern of paranasal cavities of the hairy long-nosed armadillo (*D. pilosus*) is more similar to the Southern morphotype of *D. novemcinctus* than to any other morphotype, which may have important implications on the reconstruction of its phylogenetic affinities. *Castro et al. (2015)* found this species to be the sister group of all other species attributed to the genus *Dasypus*, and therefore proposed to place it in its own genus *Cryptophractus*. This early diverging position and generic status is in disagreement with a more recent mitogenomic analysis, which retrieved *D. pilosus* in a more nested position within the

genus *Dasypus*, with *D. kappleri* representing the earliest diverging species (*Gibb et al., 2016*). Remarkably, our findings may provide new morphological arguments for such a nested position of *D. pilosus* as unambiguously supported by molecular data. The related species *D. septemcinctus* and *D. hybridus*, for their part, closely resemble each other, as it could have been expected given their overall morphological resemblance and their phylogenetic proximity. This observation adds to the growing body of evidence that these two parapatric species might in fact represent a single taxonomic entity with a large distribution (*Abba & Superina, 2010*; *Gibb et al., 2016*).

Most importantly, the variation within the nine-banded armadillo (*Dasypus novemcinctus*) allowed clearly separating three distinct geographical groups based on the pattern of paranasal cavities (Fig. 6). These individual subsets do not exactly correspond to traditional subspecies proposed for the nine-banded armadillo (*McBee & Baker, 1982*) though the distinction between the Northern and Central American (Northern morphotype) and the Southern American (Southern morphotype) groups may recall some subspecific boundaries (see below). In fact, although bone transparency often offers the possibility to observe the boundaries between the frontal sinuses and recesses, it seems that these characters have long been overlooked in cingulate systematics. The most interesting result lies in the distinction of a well-characterized entity restricted to the Guiana Shield area. Guianan nine-banded armadillos are distinguished by an inflated CFS2 in comparison to all other armadillos investigated here. The irregular outline of the CFS2 varies greatly among individuals belonging to the Guianan morphotype but its large size relative to other CFS appears distinctive. While nine-banded armadillos from the Guiana Shield have never been distinguished as a subspecies (i.e., they were until now considered as part of the subspecies *D. novemcinctus novemcintus* Linnaeus 1758; *Wetzel et al., 2008*), mitochondrial data showed that populations from French Guiana may represent an early diverging and previously unrecognized lineage clearly separated from other *D. novemcinctus* (*Gibb et al., 2016*). Specimens from French Guiana present unexpectedly distant mitochondrial D-loop region (*Huchon et al., 1999*) and divergent mitogenomes (*Gibb et al., 2016*) from the invasive US populations of nine-banded armadillos. Based on these new data, nine-banded armadillos from French Guiana are supposed to have diverged 3.7 Ma ago from a clade formed by other *D. novemcinctus*, *D. sabanicola*, *D. mazzai* and *D. pilosus* (*Gibb et al., 2016*). In this regard, the new data on paranasal cavities deliver unprecedented and very enlightening results: there exists a discrete morphological signal of internal cranial structures that supports the distinctness not only of French Guianan specimens, but also of specimens from Suriname, Guyana and the state Amapa in Brazil (Fig. 6). Based on this distribution, we refer to this entity as specimens from the Guiana Shield (or Guianan specimens) whereas we do not know the exact outline and boundaries of the range occupied by these distinctive armadillos. Taken together with recent mitogenomic data (*Gibb et al., 2016*) and analyses of cranial shape variation (*Hautier et al., 2017*), the paranasal autapomorphies found in this study make a strong case for the distinction of nine-banded armadillo specimens from the Guiana Shield as a potentially new species. The discovery of discrete paranasal characters supporting this purportedly distinct species demonstrates the necessity to study internal anatomy for a truly integrative

taxonomy. The number and delimitation of subspecies recognized within *D. novemcinctus* has long been a matter of debate among armadillo taxonomists (*Cabrera, 1958*; *McBee & Baker, 1982*; *McBee, 1999*; *Wetzel et al., 2008*; *McDonough & Loughry, 2013*). Alongside the Guianan morphotype, the study of paranasal cavities also permitted to distinguish a mostly North and Central American morphotype (Northern group) and another South American morphotype (Southern group), which largely comes from the Amazon area (Fig. 6). The Northern morphotype is characterized by (i) an anteroposterior elongation of the CFS2 to 5, with the obliquely oriented pair of CFS2 contacting each other posteromedially, and (ii) subdivided and relatively shortened RFR1. The area where this morphotype is found fully covers the proposed repartition of the subspecies *D. novemcinctus mexicanus* (*Peters, 1864*), *D. novemcinctus davisi* Russel 1953, and part of *D. novemcinctus fenestratus* (*Peters, 1864*), and *D. novemcinctus aequatorialis* (*Lönnberg, 1913*; *Wetzel et al., 2008*; *McDonough & Loughry, 2013*). It is generally well distinguished from the Southern morphotype, which is characterized by an anteroposteriorly reduced posterior chain of CFS and an elongated RFR1 (Fig. 6). The area occupied by specimens belonging to this morphotype corresponds mostly to the subspecies *D. novemcinctus novemcintus* (to the notable exception of the Guiana Shield area) and may also cover the distribution of *D. novemcinctus mexianae* Hagmann 1908 (*Wetzel et al., 2008*).

Problematic specimens whose pattern of paranasal sinuses is not easily referable to one of the three main morphotypes are present in Panama and in the eastern parts of Colombia and Venezuela (Fig. 6). This geographic area also partly corresponds to the subspecies *D. novemcinctus fenestratus* (*Wetzel et al., 2008*). The partial incongruence of these internal data with recognized subspecies of *D. novemcinctus* raises important taxonomic issues. In addition, these challenging results may also call into question the validity of the debated species *Dasypus sabanicola* (*Mondolfi, 1968*; *Abba & Superina, 2010*; *Gibb et al., 2016*), whose paratype MBUCV 439 exhibits the pattern of paranasal cavities of the *D. novemcinctus* Southern morphotype. However, this paratype represents a subadult specimen (*Mondolfi, 1968*), which casts doubts on the growth stage exhibited by its paranasal cavities (NB: other specimens attributed to this species could not be checked). The possibility exists that this morphotype represents a plesiomorphic condition within the genus, since *D. pilosus* also exhibits a similar pattern. The question as to whether or not the three *D. novemcinctus* paranasal morphotypes represent natural taxonomic entities is now to be evaluated through a thorough revision of the *Dasypus* species complex that should integrate various morphological aspects and substantial molecular data (*Hautier et al., 2017*; M-C. Arteaga, 2017, unpublished data). The case of the problematic specimens found in Colombia, Venezuela and Panama clearly illustrates this necessity.

The existence of different morphotypes within the large geographical range of *D. novemcinctus* also raises the possibility that divergent paranasal morphologies reflect adaptation to different local climatic and environmental conditions. Yet the potential functional benefits for selecting one of these paranasal patterns remain obscure. Though we cannot discard that these structures have different functions and architectural constraints, genetic drift might have also played an important role in the differentiation of these labile paranasal cavities. Geographical and environmental barriers, the Andes in particular, seem

to separate some of these cranial morphotypes (see also *Hautier et al., 2017*), a pattern that emphasizes the role played by the Andean uplift in the diversification of several xenarthrans species (*Moraes-Barros & Arteaga, 2015*).

## CONCLUSIONS

As an early worker on *Dasypus* systematics, *Hamlett* (*1939*: 335) noted that in spite of the dispersion of *D. novemcinctus* through many geographical regions, "it remains so uniform that it is apparently impossible to find external variations sufficiently constant to be of subspecific rank". In fact, he suspected that cranial characters could offer the only promise for subspecific analysis of the species. These words resonate particularly, as the strong geographical imprint found in the variation pattern of paranasal cavities sheds new light on the delimitation of *D. novemcinctus* and its subspecies. As demonstrated in this work, the investigation of frontal sinuses may help to uncover previously overlooked phylogenetic subsets within the large geographic range of nine-banded armadillos. This study highlights the under-exploited potential of internal characters for systematic studies and their utility for detecting otherwise potentially cryptic species. The strong variation and high discriminatory power found in the paranasal sinuses of armadillos is even strangely reminiscent of the extremely variable frontal sinuses of modern humans which can be used as forensic fingerprints (*Kim et al., 2013*) and kinship markers (*Szilvássy, 1982*; *Slavec, 2005*). In addition to its great potential for extant species, the study of the paranasal spaces also constitutes a promising approach to provide new informative characters for the phylogenetic placement of fossil species of the genus *Dasypus* (e.g., see partly exposed frontal sinuses in *D. punctatus*; *Castro et al., 2013*).

**Institutional abbreviations**

| | |
|---|---|
| **AMNH** | American Museum of Natural History, New York, USA |
| **BMNH** | British Museum of Natural History (Natural History Museum), London, UK |
| **IEPA** | Instituto de Pesquisas Científicas e Tecnológicas do Estado do Amapá in Macapá, Brazil |
| **KWATA** | Kwata Association collection, Cayenne, French Guiana |
| **LSU** | Louisiana State University, Baton Rouge, LA, USA |
| **MBUCV** | Museo de biología de la Universidad central de Venezuela |
| **MHNG** | Muséum d'Histoire Naturelle in Geneva, Switzerland |
| **MNHN.ZM.MO** | collections "Zoologie et Anatomie comparée, Mammifères et Oiseaux" of Muséum National d'Histoire Naturelle, Paris, France |
| **MUSM** | Museo de Historia Natural-Universidad Nacional Mayor de San Marcos, Lima, Peru |
| **NMNH** | National Museum of Natural History, Smithsonian Institution; Washington, DC, USA |
| **RMNH** | Naturalis Biodiversity Center, Leiden, Netherlands (Rijksmuseum van Natuurlijke Historie) |
| **ROM** | Royal Ontario Museum in Toronto, Canada. |

**Anatomical abbreviations and measurements**

| | |
|---|---|
| **CFS** | caudal frontal sinus (numbered from 0 to 5) |
| **FR** | frontal bone |
| **LA** | lacrimal bone |
| **LTC** | length total cranium, measured from the anterior nasal tip to the posteriormost extent of the nuchal occipital crests |
| **NA** | nasal bone |
| **NSP** | nasopharynx |
| **RFR** | rostral frontal recess (numbered from 1 to 3) |
| **RL** | lacrimal recess (numbered from 1 to 2) |
| **RMXC** | caudal maxillary recess |
| **RMXR** | rostral maxillary recess |
| **ZA** | zygomatic arch. |

# ACKNOWLEDGEMENTS

We are grateful to Géraldine Véron and Aurélie Verguin (Muséum National d'Histoire Naturelle, Paris), Roberto Portela Miguez, Louise Tomsett, Laura Balcells and Paula Jenkins (British Museum of Natural History, London), François Catzeflis and Suzanne Jiquel (Institut des Sciences de l'Evolution, Montpellier), Victor Pacheco (Dpto de Mastozoología, Museo de Historia Natural, Universidad San Marcos, Lima), Eileen Westwig (American Museum of Natural History, New-York), Burton Lim (Royal Ontario Museum, Toronto), Edmison Nicole and Chris Helgen (National Museum of Natural History, Washington), Jake Esselstyn (Louisiana State University, Bâton-Rouge), Manuel Ruedi (Muséum d'Histoire naturelle, Geneva), Claudia Regina da Silva (Instituto de Pesquisas Científicas e Tecnológicas do Estado do Amapá, Macapá), Steven van der Mije (Naturalis Biodiversity Center, Leiden), Lucile Dudoignon (KWATA association), Maria-Clara Arteaga, Maria Nazareth da Silva (Manaus Museum) and their collaborators for access to comparative material. R. Lebrun (Institut des Sciences de l'Evolution, Montpellier), Farah Ahmed (British Museum of Natural History, London), Miguel García-Sanz (Platform AST-RX MNHN) generously provided help and advice on the acquisition of μCT scans. Thanks to Sandrine Ladevèze for providing μCT-scan data for *Cabassous* and *Zaedyus*. Many thanks to Alana Sharp, Irina Ruf and an anonymous reviewer for their fruitful comments on this manuscript. In compliance with Advantages and Benefits Sharing policy in French Guiana, material from French Guiana has been registred in the collection JAGUARS (http://kwata.net/la-collection-jaguars-pour-l-etude-de-la-biodiversite.html; CITES reference: FR973A) supported by Kwata NGO, Institut Pasteur de la Guyane, DEAL Guyane, and Collectivité Territoriale de la Guyane. This is contribution ISEM 2017-127 of the Institut des Sciences de l'Evolution.

### Funding

This work has benefited from an "Investissements d'Avenir' grant managed by Agence Nationale de la Recherche, France (CEBA, ref. ANR-10-LABX-25-01). This research received support from the Synthesys Project (http://synthesys3.myspecies.info/), which is financed by the European Community Research Infrastructure Action under the FP7. The funders had no role in study design, data collection and analysis, decision to publish, or preparation of the manuscript.

### Grant Disclosures

The following grant information was disclosed by the authors:
Agence Nationale de la Recherche.
European Community Research Infrastructure Action.

### Competing Interests

Benoit de Thoisy is an employee of Association Kwata, Cayenne, French Guiana. Otherwise, the authors declare that they have no competing interests.

### Author Contributions

- Guillaume Billet conceived and designed the experiments, performed the experiments, analyzed the data, contributed reagents/materials/analysis tools, wrote the paper, prepared figures and/or tables, reviewed drafts of the paper.
- Lionel Hautier conceived and designed the experiments, performed the experiments, contributed reagents/materials/analysis tools, reviewed drafts of the paper.
- Benoit de Thoisy contributed reagents/materials/analysis tools, reviewed drafts of the paper.
- Frédéric Delsuc conceived and designed the experiments, contributed reagents/materials/analysis tools, reviewed drafts of the paper.

### Data Availability

The raw data has been provided as a Supplemental File.

### Supplemental Information

Supplemental information for this article can be found online at http://dx.doi.org/10.7717/peerj.3593#supplemental-information.

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
