# Peer review of "The hidden anatomy of paranasal sinuses reveals biogeographically distinct morphotypes in the nine-banded armadillo (Dasypus novemcinctus)"

_PeerJ, doi:10.7717/peerj.3593_

## Round 0.1 · original submission · Minor Revisions

· Academic Editor

Minor Revisions

Dear authors,

I have received three reviews for your manuscript. All reviewers suggest that this is a great and well-presented study that will represent a valuable addition to the literature, and which should be published following some minor revisions. Please address these minor comments, particularly reviewer #1 has suggested an additional discussion point and reviewer #3 has requested some additional method-related detail. All the reviewers have noted that there are a few grammatical issues here and there in the text, and some typos - please take care to read through the text and correct these.

I look forward to receiving your revised manuscript.

·

Basic reporting

The paper is clear and well written; a few minor typos and suggestions are pointed out below:
Line 59 – uncover, not uncovering
Line 121 – through, not throughout
Line 245, 284, 315 – also reference figure 6 in the results
Line 399 – remove “of”
Line 446 – add “of” in “…, but the size and shape of frontal sinuses…”
Figure 6 caption – remove "of the" in "Each of the morphotype is represented by…"
Figure 4 caption – state that they are reconstructions of the adult specimens.

The figures are excellent.

This isn’t necessary but I’d recommend adding my paper on the mechanical role and mass reducing effect of large sinuses in Diprotodon (Sharp AC, Rich TH (2016) Cranial biomechanics, bite force and function of the endocranial sinuses in Diprotodon optatum, the largest known marsupial. Journal of Anatomy 228: 984-995.) at either line 428, 429 or 432.

Experimental design

no comment

Validity of the findings

no comment

Additional comments

This is a very interesting study demonstrating the importance of using concealed morphological differences to detect variation within and between species. The aims and conclusions are clear, and the methods and results are sufficiently detailed. As far as I am concerned, the paper could be published as is, with just a few very minor suggestions.

A discussion on why there are different morphotypes within D. novemcinctus over the geographical range would be interesting, especially as the effects of climate and biogeographic events are mentioned in the abstract but not discussed. I imagine this might be discussed more in the accompanying paper (Hautier et al. unpublished data) which is on cranial shape variation, however, if not it would be good to include a brief discussion.

Reviewer 2 ·

Basic reporting

Overall, I was able to understand the language used in the manuscript. However, there were minor grammatical issues throughout that could easily be resolved by having a native English speaker edit the manuscript. In several instances, as indicated in the attached pdf, the authors should use more appropriate citations, but overall they provide sufficient background and context fore their paper. The structure of the article, figures, and tables are all appropriate, and they shared their raw data. The results are very thorough and relevant.

Experimental design

I believe that this work is very valuable and adds to our understanding of internal cranial morphology and its importance in systematics. The authors used large sample sizes, which are crucial in studies of variable structures such as paranasal sinuses. In several cases, as indicated in the attached pdf, more details about data collection would be helpful for replication.

Validity of the findings

The findings are clearly illustrated and described and I believe that the results presented are sound.

Additional comments

In their manuscript, "The hidden anatomy of paranasal sinuses reveals biogeographically distinct morphotypes in the nine-banded armadillos (Dasypus novemcinctus)" Billet and colleagues explored whether internal cranial anatomy, specifically paranasal sinuses and recesses, can be used to distinguish among different subspecies of nine-banded armadillos. The authors provide solid evidence that paranasal sinus morphology is distinct among subspecies, despite external homogeneity in skull morphology. I felt that this paper was interesting, but did have several minor areas that could be improved. First, the manuscript would benefit from being read for correct English grammar. However, I was able to understand the manuscript. I also felt that the literature cited was not always appropriate, which I indicated in the attached annotated pdf, and some parts of the methods section should be more detailed. Overall, the paper was very interesting and with some minor revisions would make a great addition to the anatomical literature.

Annotated reviews are not available for download in order to protect the identity of reviewers who chose to remain anonymous.

·

Basic reporting

1) The use of terms concerning the µCT scans should be consistent throughout the manuscript (see comments in the PDF).

2) The topic is presented very well and the reference list covers the most important literature related to the field of research. However, there are two references missing that cover prenatal ontogeny of Dasypus novemcinctus and Zaedus minutus, and might be important for the discussion:
REINBACH, W. (1952): Zur Entwicklung des Primordialcraniums von Dasypus novemcinctus LINNÉ (Tatusia novemcincta LESSON).- Teil 1:Z. Morph. Anthrop. 44: 375-444.- Teil 2: Z. Morph. Anthrop. 45: 1-72.
REINBACH, W. (1955): Das Cranium eines Embryos des Gürteltieres Zaedus minutus (65 mm Sch.-St.).- Morph.Jb. 95: 79-141.

3) Some figures could be improved by additional labeling (see comments in the PDF)

4) I missed the figure capture of Fig. S1.

Experimental design

1) The identification of sinuses and recesses just by external investigation (based on photographs) should be explained in more detail.

Validity of the findings

no comment

Additional comments

Great work and well written manuscript!
Please see comments and corrections in the attached PDF.

---

## Round 0.2 · accepted · Accept

· Academic Editor

Accept

Dear authors,

Thank you for addressing the earlier, minor comments to your manuscript, I think your revised version is now ready for publication.